# Investigating Friction and Antiwear Characteristics of Organic and Synthetic Oils Using h-BN Nanoparticle Additives: A Tribological Study

**Umida Ziyamukhamedova [1], Shah Wasil [2], Sanjay Kumar [2], Rakesh Sehgal [2], M. F. Wani [2,\*], Chandra Shekhar Singh [2], Nodirjon Tursunov [1] and Himanshu Shekhar Gupta [2]**

[1] Department of Material Science and Mechanical Engineering, Faculty of Railway Engineering, Tashkent State Transport University, Tashkent 100067, Uzbekistan; z.umida1973@yandex.ru (U.Z.)

[2] Mechanical Engineering Department, National Institute of Technology Srinagar, Hazratbal, (J&K), Srinagar 190006, India; sanjay_03phd19@nitsri.ac.in (S.K.); rakeshsehgal.nitham@gmail.com (R.S.)

\* Correspondence: mfwani@nitsri.ac.in

**Abstract:** Friction and wear are two major elements that influence the life of a variety of equipment. According to estimates, as much as 30% of the energy used is dissipated as friction. However, this figure can be reduced by developing materials that enhance surfaces and apply lubricants appropriately. This study aims to analyze the impact of hexagonal boron nitride (h-BN) nanoparticles on the rheological and antiwear characteristics of four distinct oil varieties, namely Society of Automotive Engineers SAE-20W50, soybean, Polyalphaolefin PAO-4, and olive oils. The results of the tribological tests demonstrated a noteworthy enhancement in antiwear characteristics with the addition of 0.2 wt.% of h-BN nanolubricant. Among all nanolubricants, the maximum reduction in coefficient of friction (COF) and wear scar diameter was observed at 0.2 wt.% of h-BN in SAE-20W50 oil. Compared to the base oils, the wear scar diameter decreased by 17.93%, 8.80%, 22.65%, and 24.04% for (soybean oil and 0.2 wt.% of h-BN), (SAE20W50 and 0.2 wt.% of h-BN), (PAO4 and 0.2 wt.% of h-BN), and (olive oil and 0.2 wt.% of h-BN), respectively. Rheological test results indicated that with the addition of h-BN nanoparticles, the viscosity of the base oils significantly increased. The maximum viscosity was observed at 40 °C for SAE20W50 base nanolubricants, according to the rheological measurements.

**Keywords:** sustainability; nanolubricants; antiwear; rheology

## 1. Introduction

With increased awareness of the importance of environmental preservation, sustainable and biodegradable lubricants have gained popularity. In addition to being non-toxic, biodegradable oils possess low volatility due to the high molecular weight of the triglyceride molecule [1–4]. Nonetheless, the primary challenge in using biodegradable base stocks as a next-generation lubricant lies in surpassing the performance of mineral oils in terms of presenting desirable properties. Biolubricants are often associated with drawbacks such as inferior oxidative stability, a high pour point, and a high coefficient of friction (COF) at elevated temperatures [5,6]. The superior lubricity of vegetable oils relative to mineral-based oils can be attributed to their surface adsorption abilities and the formation of a polar head layer that adheres to the surface [7]. Some studies show that vegetable oils have excellent friction and wear characteristics. Such properties have been identified in many vegetable oils, such as soybean oil, sunflower oil, olive oil, rapeseed oil, coconut oil, garlic oil, etc. Lubricants, in their purest form, are not as effective at considerably reducing friction and wear [7].

Recently, nanoparticles (NPs) have been studied as additives in lubricants due to their wear and friction reduction characteristics. Broadly, these NPs, under nano-scale boundary lubrication regimes, have the advantage of easily entering asperities and acting

as micro-bearings and/or forming tribe-films, thereby reducing the friction and, hence, the wear in the mating surfaces. Such effects are studied in many NPs, like tungsten sulphide (WS2), molybdenum di-sulphide ($MoS_2$), hexagonal boron nitride (h-BN), aluminum oxide ($Al_2O_3$), titanium oxide ($TiO_2$), copper oxide (CuO), graphene nanoplatelets (GNP), etc. On the other hand, h-BN nanoparticles also show excellent friction and wear reduction properties. Hexagonal boron nitride (h-BN) is a highly stable crystalline form of boron nitride that exhibits a layered structure similar to that of graphite and is commonly known as "white graphite". The layers within h-BN are bound by strong covalent bonds between boron and nitrogen atoms, while the layers themselves are held together by weak van der Waals forces. The interlayer arrangement of these sheets differs from that of graphite because of the eclipsed arrangement of boron atoms over nitrogen atoms [8]. This unique structure makes h-BN an ideal material for lubrication, as it exhibits a "ball bearing effect" and a "nano polishing effect" that are effective in reducing friction and wear. Furthermore, h-BN has a layered structure with hexagonal rings composed of sp2 hybridization and possesses weak van der Waals forces, making it well suited for lubrication applications. [8]. Numerous research studies have investigated the influence of h-BN nanoparticle additives on the tribological performance of sliding components. These investigations involved the addition of h-BN NPs to various types of oils, such as commercial engine oil (SAE-20W40, SAE-20W50, etc.) [9,10], mineral oil [11], and organic oil [12,13]. The outstanding ability of h-BN NPs to reduce friction and wear has also been extensively explored. Various studies have suggested distinct mechanisms, such as the mending [9] and ball-bearing mechanisms [14,15], for h-BN NPs. These nanoparticles have been utilized as additives in engine oil, as well as in polyalphaolefin and jatropha oil.

Abdullah et al. [16] investigated the friction wear properties of engine oil SAE 15W40 by incorporating 0.5 vol.% of h-BN NPs and oleic acid (0.3 vol.%) as a surfactant. They observed that a thin fluid film forms between the contact interfaces, which helps to enhance the antiwear properties of SAE 15W40. Charoo et al. [17] investigated the tribological properties of h-BN NPs as lubricant additives in base oil (SAE 20W50) under different loading conditions. According to their observations, the minimum value of COF (0.0401) was observed at 3 wt.% of h-BN under a normal load of 100 N, and the minimum wear volume (0.011 $mm^3$) was observed at a normal load of 50 N. Wan et al. [18] prepared nanolubricant by incorporating boron nitride (BN) NPs as additives and investigated their tribological characteristics. Based on the experimental data, they reported that at 1 wt.% of BN, NPs notably enhance the anti-friction and antiwear characteristics of the base oil. Thorat et al. [19] investigated the tribological properties of SAE 10W30 lubricant using hexagonal BN NPs. According to their observations, h-BN NPs play a significant role in enhancing the antiwear properties of the base oil. The optimum result was achieved using a 0.5% h-BN blend with SAE 10W30 base oil, which demonstrated the lowest friction coefficient. Kerni et al. [20] conducted a study that aimed to improve the oxidation stability of olive oil and investigate the impact of varying concentrations of h-BN NPs on the wear and friction characteristics. The authors reported that the lubricating characteristics of epoxidized oil significantly improve in comparison to pure olive oil. Ilie et al. [21] investigated the tribological properties of titanium oxide ($TiO_2$)-based nanolubricant by employing Fourball and ball-on-disc tribotests. The authors reported that the prepared nanolubricants exhibited excellent dispersion stability. Moreover, the friction coefficient and antiwear properties were significantly improved via use of $TiO_2$ as a lubricant additive. Gupta et al. [22] investigated the rheological and tribological characteristics of Mahua oil and Flaxseed oil by incorporating the $TiO_2$, $ZrO_2$, Graphite, and $MoS_2$ nanoparticles as lubricant additives. They confirmed that the $TiO_2$ and graphite nano-additives provided superior antiwear properties in Mahua oil, with the maximum reduction in wear scar diameter being 6.5% and 10.14% at an optimum concentration of nano-additives of 0.75 wt.%. In contrast, $TiO_2$, $ZrO_2$, and $MoS_2$ nano-additives had superior antiwear properties than Flaxseed oil, with the maximum reductions in wear scar diameter being 30.2%, 11.7%, and 15% at optimum concentrations of 0.25 wt.%, 0.75 wt.%, and 1 wt.%, respectively. Gupta et al. [23] formu-

lated the nanolubricants of Mahua oil by introducing GNP, ZnO, and CuO nanoparticles. They reported that the addition of these nanoparticles into Mahua oil provided outstanding antifriction characteristics. The maximum reductions in COF were 6.27%, 22%, and 68.8% with optimal concentrations of 0.5 wt.% (GNP and ZnO nanoparticles) and 1 wt.% (CuO nanoparticles), respectively. They also confirmed that the Mahua oil has anti-corrosion characteristics superior to those of 20W40 commercial oil, and the incorporation of these NPs had no significant impact on the corrosion properties of mahua oil.

The objective of this study is to identify the most appropriate base stock oil that exhibits low COF and antiwear properties. Currently, there are insufficient studies of the incorporation of h-BN NPs into both organic and synthetic oils. Therefore, it is crucial to investigate these novel aspects to determine the feasibility of developing nanolubricants for various engineering applications. This study examines and compares the tribological properties of soybean oil, olive oil, PAO oil, and SAE-20W50 that contain 0.2 wt.% of h-BN NPs. The experiments were conducted using a four-ball tribotester, while the rheological properties and dynamic viscosity of both the base oils and nanolubricants were measured using a modular compact rheometer.

## 2. Materials and Methods

### 2.1. Materials and Preparation of Nanolubricants

For this study, four base oils were chosen, namely Soybean oil, SAE-20W50 engine oil, PAO4 oil, and olive oil. The h-BN NPs (APS-25 nm) used were obtained from Nano-Shell, India, while the base oils were procured from a local vendor. The rheometer was used to determine the viscosity and shear stress properties of the normal oils at 40 °C and 100 °C, and the results are presented in Table 1.

**Table 1.** Properties of base oils.

| Oil | Viscosity (m. Pas) at 40 °C | Viscosity (m. Pas) at 100 °C | Sp. Gravity |
|:---:|:---:|:---:|:---:|
| SAE 20W50 | 140.40 | 16.6 | 0.890 |
| Soybean Oil | 31.891 | 7.886 | 0.917 |
| PAO4 | 15.30 | 3.443 | 0.68 |
| Olive Oil | 36.40 | 8.05 | 0.913 |

To achieve homogenous and stable mixtures, an ultrasonicator (WENSAR ultrasonicator) was employed to mix NPs with oil at a frequency of 40 kHz for 8 h. Nanolubricants were prepared by adding h-BN NPs (0.2 wt.%) to the base oils. Figure 1 displays the image of the prepared nanolubricants. To prepare the nanolubricants, the weight percentage of the nanoparticles was calculated using Equations (1) and (2).

$$a \ wt.\% = \frac{wt. \ of \ solute \ (x)}{wt. \ of \ solution \ (x+y)} \ \dots \tag{1}$$

$$x = \frac{ay}{100 - a} \ \dots \tag{2}$$

where x = weight of solute, y = weight of solvent, and a = weight percentage of concentration. Using the equation, the respective weights of the solute and h-BN were calculated to prepare the solutions of different oils. In total, 75 mL of oil was taken for each solution.

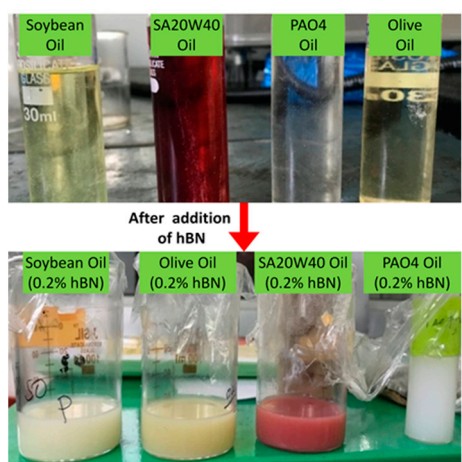

**Figure 1.** Image of the prepared nanolubricants.

## 2.2. Characterization of Nanoparticles

The crystallinity of h-BN NPs was examined via powder X-ray diffraction (XRD, Rigaku-Smart lab, Japan), with a Cu-Kα radiation source of λ = 1.54 Å and scan rate of 5 deg./min used, as shown in Figure 2a. The XRD pattern at 2θ values corresponded to the crystallographic planes (002), (100), (101), (004), and (110) of hexagonal boron nitride [24]. The d-spacing value of the nanoparticle peak observed at 2θ of 26.67° was 0.33 nm, indicating a good crystallinity of h-BN NPs. The morphological study of h-BN NPs was analyzed via Field emission scanning electron microscopy (FE-SEM-ZEISS-500, Germany), as shown in Figure 2b. A round sheet-like morphology was observed, which indicates the 2D structure of h-BN NPs. XRD and FESEM results confirm the authenticity of the purchased particles.

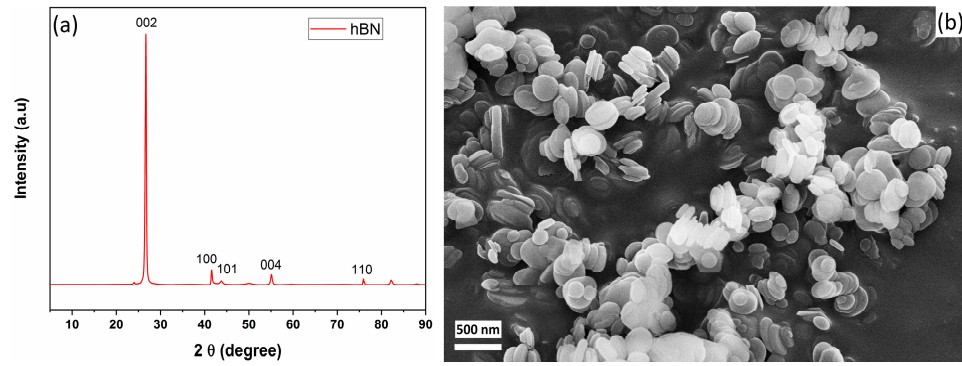

**Figure 2.** (**a**) XRD patterns of h-BN; (**b**) FESEM of h-BN NPs.

## 3. Experimental Procedure

The Anton Paar MCR102 was utilized to measure the viscosity of both base oils and nanolubricants. Figure 3a,b displays the various components of the MCR102 that are responsible for ensuring accurate viscosity measurements, such as the normal force sensor, air bearings, and EC motor. The viscosity of a lubricant was measured using cone and plate (CP-40, cone angle of 1°) geometry and radial air bearings. A flow curve test was performed at a shear rate in the range 1–600 s$^{-1}$ at different temperatures (40 °C and 100 °C). The changes in the viscosities of the prepared nanolubricants and the base oils were also assessed. In order to assess the antiwear and COF characteristics of prepared nanolubricants and base oils, a four-ball tribometer (Ducom) was used, and all tests were conducted in accordance with the ASTM-D4172 standard. As per the ASTM-D4172 standard, the tribopairs for the testing consisted of four steel balls made of AISI E52100. These steel balls each had a hardness of 60 HRC (Rockwell hardness scale) and a diameter of 12.7 mm. However, during antiwear testing, three of the steel balls were placed inside a pot or

container that was filled with the test lubricant. The fourth steel ball was attached to a spindle, creating a three-point contact configuration, as depicted in Figure 4. This setup allowed the evaluation of the lubricating properties and wear characteristics of the prepared lubricant under controlled conditions. The three-point contact configuration facilitated the measurement of frictional torque and wear rates between the rotating steel ball and the stationary steel balls immersed in the lubricant [25]. This testing method provided valuable insights into the performance of the lubricant in reducing friction and preventing wear in a simulated tribological environment. The test parameters for different samples of nanolubricants are listed in Table 2. The ASTM-D4172 standard was followed to assess the average wear scar on the three lower balls of the four-ball tester for the tested samples.

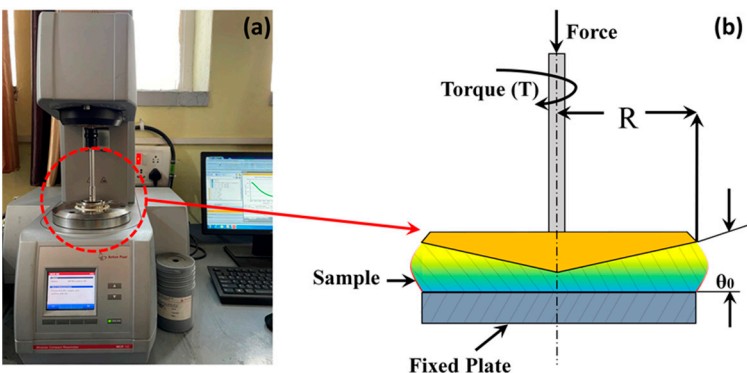

**Figure 3.** (**a**) Image of the rheometer instrument; (**b**) schematic of the cone and plate geometry.

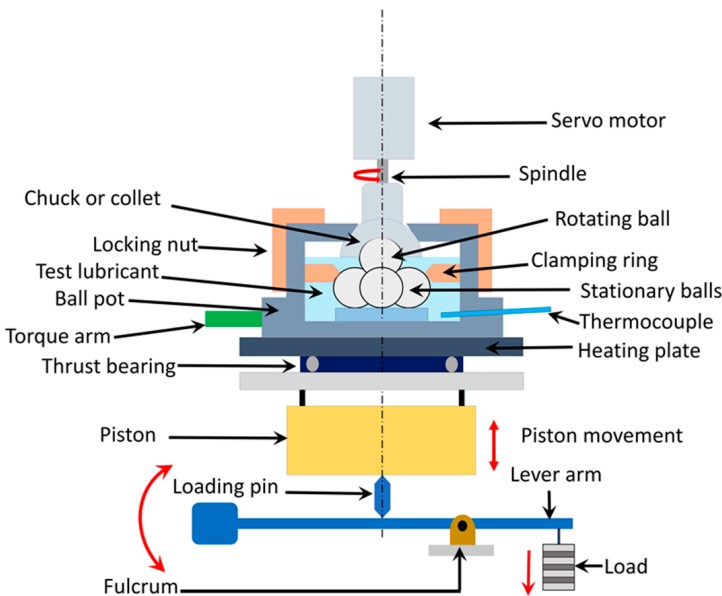

**Figure 4.** Schematic of the four-ball tribo-tester.

**Table 2.** Different samples of nanolubricants.

| S. No | Test Parameters | Test Sample |
|---|---|---|
| 01 | | Soybean Oil |
| 02 | | Soybean Oil and 2 wt.% of h-BN |
| 03 | | SAE 20W50 oil |
| 04 | Load—392 N | SAE 20W50 and 2 wt.% of h-BN |
| 05 | Speed—1200 rpm | PAO4 oil |
| 06 | Temperature—75 °C | PAO4 and 2 wt.% of h-BN |
| 07 | | Olive oil |
| 08 | | Olive oil and 2 wt.% of h-BN |

## 4. Results and Discussion

### 4.1. Rheological Measurements of Nanolubricants

The variations in shear viscosity with the shear rate of both base oils and nanolubricants at different temperatures are illustrated in Figure 5a–d. It is noteworthy that SAE20W50 exhibits the highest viscosity, while PAO-4 shows the lowest viscosity. The viscosity of olive oil is marginally increased compared to soybean oil due to the presence of a large amount of fatty unsaturated acid. At low temperatures (40 °C), the shear viscosities of soybean oil, SAE20W50, PAO-4, and olive oil are 31.891 mPa·s, 140.40 mPa·s, 15.30 mPa·s, and mPa·s 36.40, respectively. The correlation between the applied shear rate and dynamic viscosity shows a consistent pattern, which confirms the Newtonian characteristics of both base fluids and nanofluids. The reason for this outcome is that when the shear rate is increased, the molecules of the oil align themselves, and the maximum structure of the oil molecules is attained. This process leads to a steady relationship between viscosity and shear rate [26,27]. The flow behavior of the base oils is hardly affected by the incorporation of h-BN NPs. Nonetheless, the incorporation of NPs leads to a rise in the shear viscosity of all oils at a concentration of 0.2 wt.%. For soybean oil, SAE20W50, PAO-4, and olive oil, the viscosity increases are 2.427%, 0.441%, 3.594%, and 1.8131%, respectively, at 40 °C and a h-BN concentration of 0.2 wt.%. The rise in shear viscosity is due to the NPs settling between the oil layers, resulting in increased resistance being offered by the oil. In addition, the increase in dynamic viscosity can be attributed to nanoparticle agglomeration, which leads to larger particle clusters and higher flow resistance in the oil. By examining Figure 5a–d, it can be noted that the shear viscosities of both base oils and nanofluids decrease as the temperature rises. The decreases in viscosity are caused by the weakening of intermolecular forces at higher temperatures.

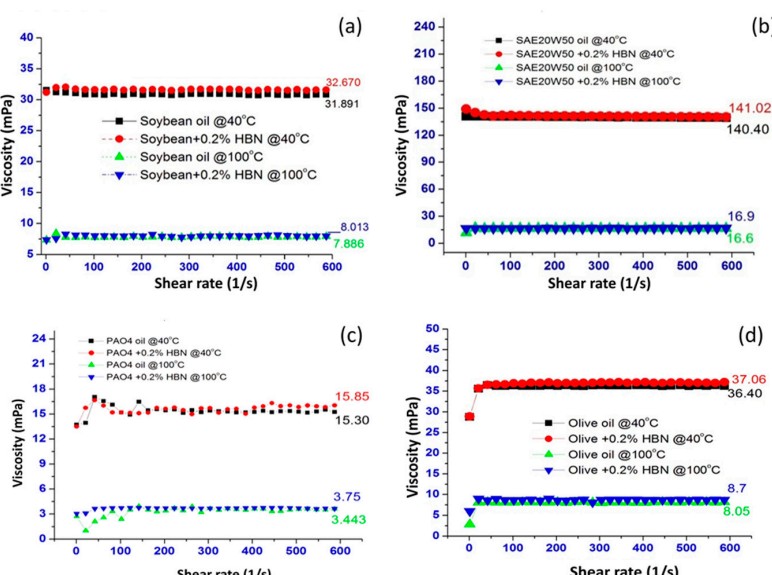

**Figure 5.** Flow curve (viscosity Vs. shear rate) of base oils and 0.2 wt.% of h-BN NPs at 40 °C and 100 °C. (**a**) soybean oil (**b**) SAE 20W50, (**c**) PAO-4, (**d**) olive oil.

### 4.2. Tribological Investigation

Antiwear Properties

The COF values for various nanofluid samples were evaluated by analyzing the data obtained via the antiwear test, and the value of the COF was calculated using Equation (3). The variations in friction torque and COF for 0.2 wt.% of h-BN NPs in different base oils are illustrated in Figures 6 and 7. In order to find the best sequence of nanolubricants that results in lower COF, a comparison was made between the base oils. The results depicted in Figure 7 show that the addition of h-BN NPs to the base oils significantly reduced the COF and friction torque [28]. The presence of 0.2 wt.% of h-BN NPs in SAE-20W50 showed

a significant reduction in the COF and frictional torque. However, the 0.2 wt.% of h-BN NPs in PAO-4 exhibit the highest COF and friction torque. Vegetable oil, soybean oil, and olive oil showed excellent performance in terms of COF and frictional torque. Olive oil with 0.2 wt.% of h-BN NPs had a lower COF and friction torque than soybean oil (0.2 wt.% of h-BN). The reason for this outcome is that the NPs reduce friction between rubbing surfaces and enhance the oil's load-carrying ability. Compared to the base oils, the friction coefficient was reduced by 4.85%, 6.89%, 7.24%, and 7.715% for Soybean and 0.2 wt.% of h-BN, 20W50 and 0.2 wt.% of h-BN, PAO4 and 0.2 wt.% of h-BN, and Olive and 0.2 wt.% of h-BN, respectively. The addition of h-BN NPs noticeably enhanced the anti-friction performance of the base oils.

$$\mu = \frac{T\sqrt{6}}{3Wr}\cdots \tag{3}$$

where $\mu$ is the COF, T is the torque of the upper shaft (kg-mm), r = 3.6 mm (distance from the axis of rotation to the center of the contact of the fixed ball), and W is the load (kg).

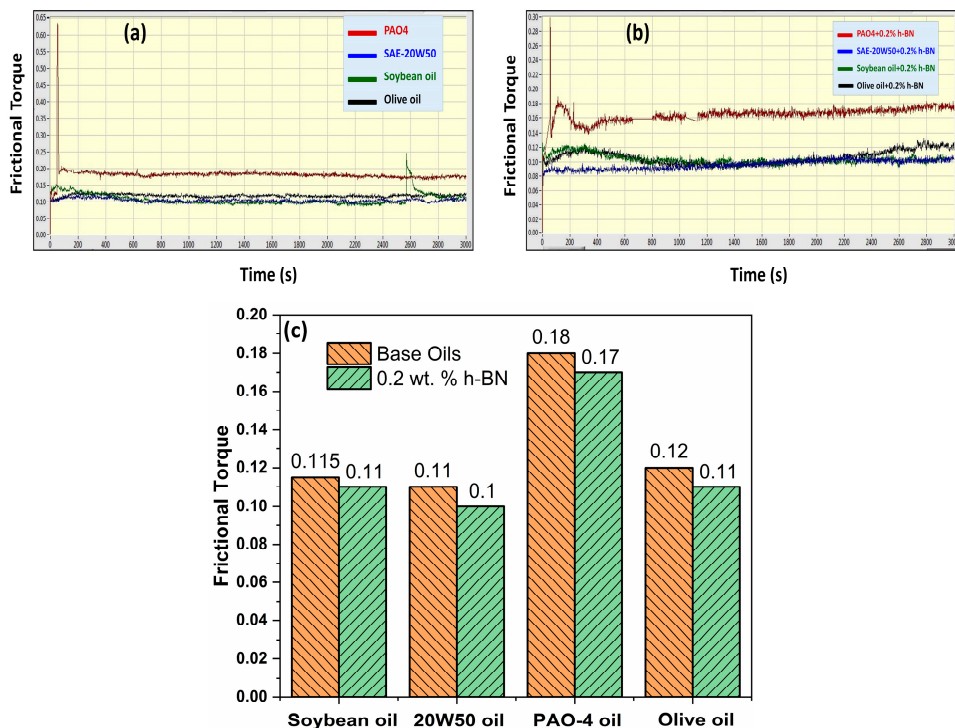

**Figure 6.** Frictional torque vs. time for (**a**) normal oils and (**b**) oils with 0.2 wt.% of h-BN NPs; (**c**) bar graph representing friction torque vs. time.

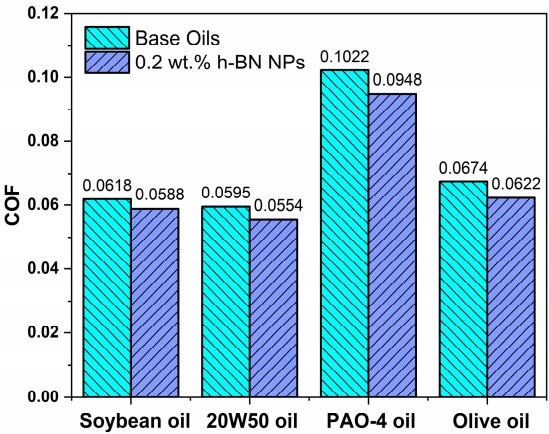

**Figure 7.** Bar graph comparing COF base oils and oils with addition of h-BN NPs.

The size of each of the wear scars for the three balls used in each sample was measured after completing the analysis of COF. The wear scars of worn-out balls were examined using an optical microscope (Leica Model: DM2500, Germany), and the wear scar diameters of the three stationary balls are listed in Table 3. Table 3 demonstrates a noteworthy alteration to the wear scar diameter across the tests conducted using h-BN NPs-based nanolubricants. Figure 8 illustrates the changes in the scar diameter when 0.2 wt.% of nanoparticles (NPs) was introduced into the base oils. Significant reductions in average wear scar diameter (AWSD) were observed for nanolubricants compared to based oils, as shown in Figures 9 and 10. The minimum AWSD (383.829 μm) was observed at SAE-20W50 and 0.2 wt.% of h-BN nanolubricant, and the maximum scar diameter (684.457 μm) was observed at olive and 0.2 wt.% of h-BN nanolubricant. However, in the case of organic oils, soybean and 0.2 wt.% of h-BN nanolubricant exhibits minimum AWSD (676.559 μm). This antiwear test revealed that h-BN had a substantial impact in terms of enhancing the wear-resistant properties of the base oils. This result can be attributed to the formation of a protective layer of h-BN NPs on the contact interface, which promotes easy shearing, as shown in Figure 11a–c. Figure 11a shows the mechanism of dry sliding condition in which more asperity-to-asperity contacts occurred between the contact interface, resulting in an increase in the COF and wear. Additionally, in the presence of lubricating sliding conditions, the asperity-to-asperity contacts in the contact interface decreased due to the formation of lubricating film, as depicted in Figure 11b. Using h-BN NPs as an additive in the base oil forms a protective film over the contact interface, which promotes easy sharing and reduces the COF and wear significantly, as illustrated in Figure 11c. Moreover, h-BN NPs serve as healing agents for surfaces with cracks, micro-holes, and porosities, which is a key factor that contributes to the decrease in the diameter of the wear scar. However, in the base oil's lubrication condition, rough and bland wear scars were observed. On the other hand, for the oil with h-BN NPs, a smoother wear scar was observed, and this observation was further confirmed via an enlarged view of the wear scar, as shown in Figure 12.

**Table 3.** Average wear scar diameter (measured in μm) for tested samples.

| Oil/Average Wear Scar Diameter (μm) | Ball 1 | Ball 2 | Ball 3 | AWSD (μm) |
|---|---|---|---|---|
| Soybean Oil | 900.96 | 712.290 | 859.77 | 824.341 |
| Soybean Oil and 2 wt.% of h-BN | 719.168 | 614.354 | 696.156 | 676.559 |
| SAE 20W50 oil | 425.369 | 418.672 | 418.513 | 420.904 |
| SAE 20W50 and 0.2 wt.% of h-BN | 376.980 | 387.253 | 387.260 | 383.829 |
| PAO4 oil | 781.096 | 791.353 | 753.205 | 775.24 |
| PAO4 and 2 wt.% of h-BN | 605.516 | 608.017 | 585.26 | 599.603 |
| Olive oil | 931.90 | 881.452 | 889.756 | 901.039 |
| Olive oil and 0.2 wt.% of h-BN | 664.835 | 723.70 | 664.835 | 684.457 |

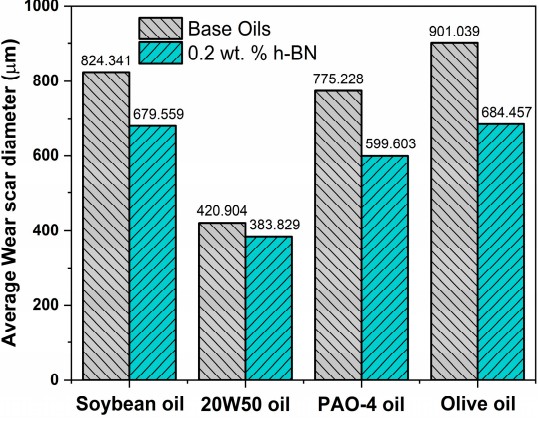

**Figure 8.** Comparison between wear scar diameters of base oils and nanolubricants.

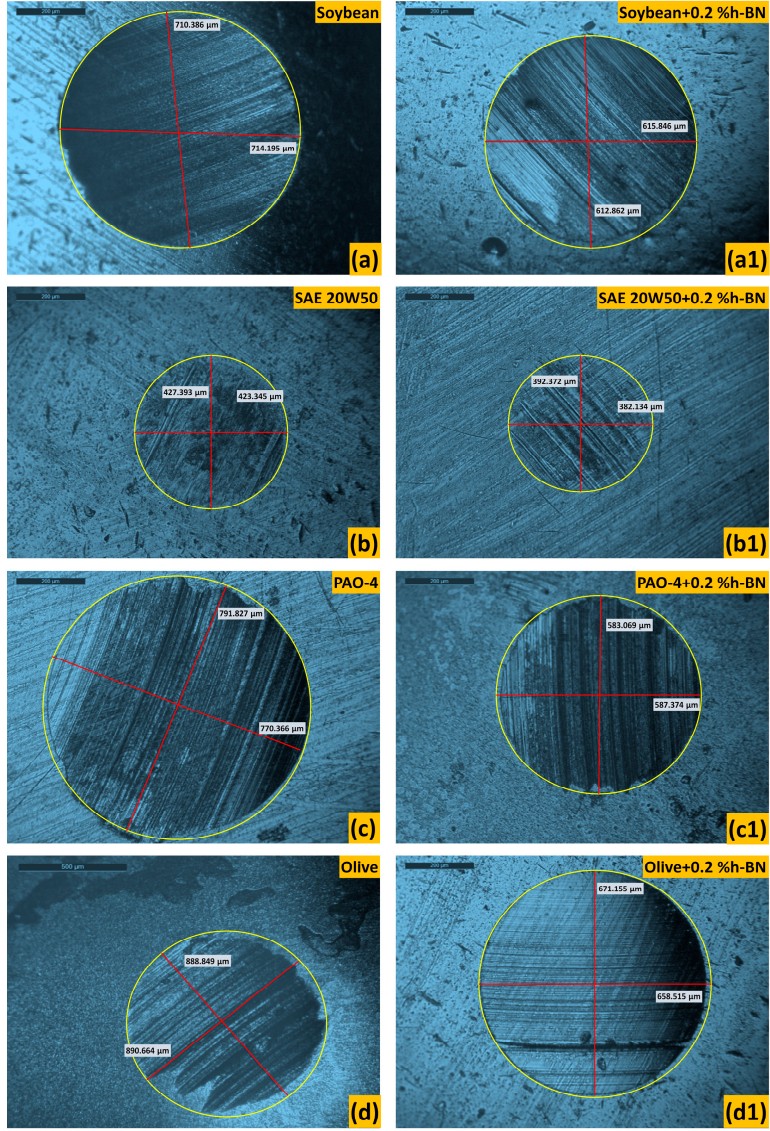

**Figure 9.** Optical images of wear scars for one ball derived from each sample: (**a–d**) oil without nanoparticles and (**a1–d1**) oil with nanoparticles (0.2 wt.% of h-BN).

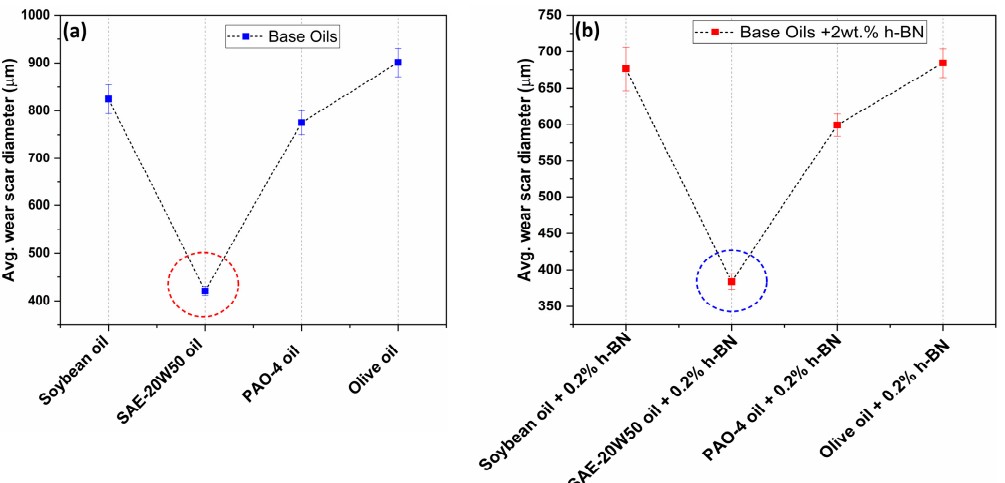

**Figure 10.** The mean diameter of the wear scar (**a**) without NPs or (**b**) with h-BN NPs.

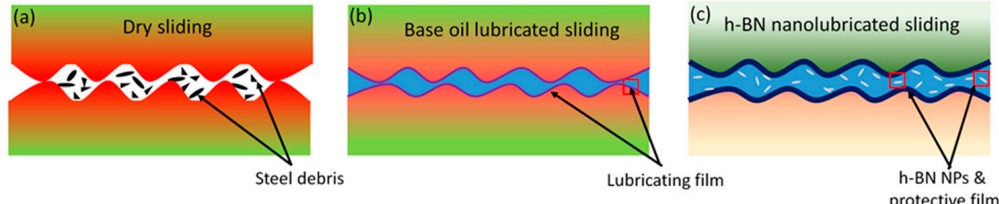

**Figure 11.** Schematic of (**a**) dry, (**b**) base oil-lubricated, and (**c**) h-BN nano-lubricated sliding conditions.

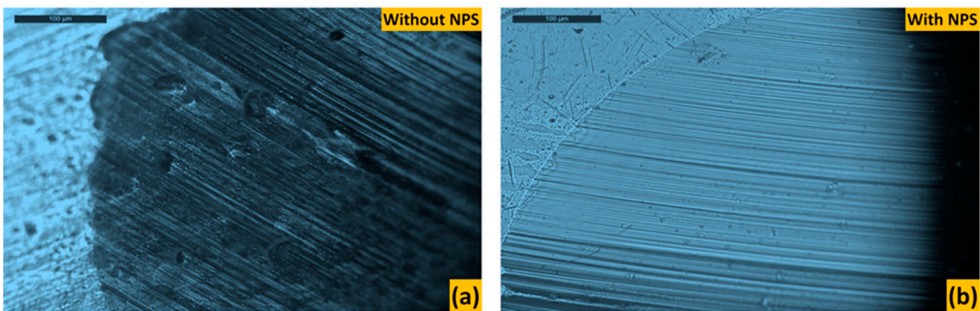

**Figure 12.** Enlarged optical images of wear for (**a**) normal oil without h-BN NPs and (**b**) oil with 0.2 wt.% h-BN NPs.

The worn-out surface morphology of wear scars was determined via FE-SEM, as depicted in Figure 13a,b. Under base oil-based lubricating conditions, the worn-out surface exhibited significant abrasive marks with some grooves and micro-crack formation, as shown in Figure 13a. Generally, wear scars obtained under base oil-based lubricating conditions displayed a rough appearance, with noticeable material removal occurring. The abrasion marks were prominent, large, and deep, indicating significant wear loss. In contrast, under nanolubricated conditions, a remarkable reduction in the intensity of abrasion marks was observed, and the wear scars exhibited smoother surfaces, as illustrated in Figure 13b. The presence of a protective film formed via h-BN NPs on the contact surfaces is the possible reason for the significant decrease in the diameter of the wear scar [9]. This protective film acts as a barrier, minimizing direct contact between the surfaces and reducing direct metal-to-metal contact. Consequently, the wear scars obtained via nanolubrication exhibited less wear than those obtained using base oils.

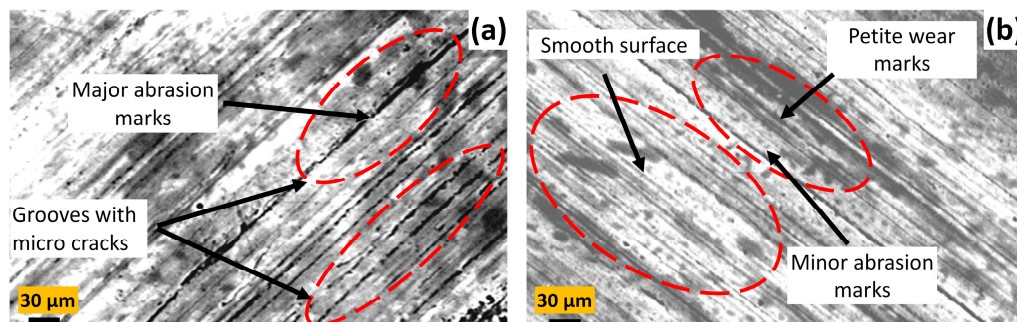

**Figure 13.** SEM Image of wear scar: (**a**) normal oil without h-BN NPs; (**b**) oil with h-BN NPs of 0.2 wt.%.

## 5. Conclusions

In the present study, the lubrication behaviors of organic and synthetic oils were investigated both with and without the use of h-BN NPs. The rheological and tribological properties of base oils and h-BN NP nanolubricants were investigated. The major findings of this investigation are as follows:

1. The rheological test results indicated that the viscosity of the prepared nanolubricants improved at both 40 °C and 100 °C compared to the viscosity of the base oils. However, among the synthetic oils, PAO and 0.2 wt.% of h-BN exhibited the highest increase in viscosity (3.594%) compared to the SAE20W50 and 0.2 wt.% of h-BN nanolubricant. Moreover, within the organic oils, the maximum increase in viscosity was observed for the soybean oil and 0.2 wt.% of h-BN when compared to the olive oil and 0.2 wt.% of h-BN nanolubricant.

2. The addition of h-BN NPs resulted in a significant reduction in the COF compared to that of the base oils. The antiwear test results indicated that the maximum reduction in COF was observed in the organic oil relative to the results of the synthetic oil. However, among the nanolubricants, the lowest COF was observed for soybean oil and 0.2 wt.% of h-BN, while the maximum COF was observed for PAO and 0.2 wt.% of h-BN nanolubricants.

3. The incorporation of h-BN NPs was very beneficial in terms of enhancing the antiwear properties of base oils. Among the synthetic oils, the minimum wear scar diameter was observed for SA20W50 and 0.2 wt.% of h-BN nanolubricants compared to PAO-4 and 0.2 wt.% of h-BN nanolubricants. However, in the case of organic oils, the minimum wear scar diameter was observed for soybean and 0.2 wt.% of h-BN compared to olive oil and 0.2 wt.% of h-BN nanolubricants.

4. The abrasion wear mechanism with crack and groove formation was observed under base oil-based lubricating conditions. In contrast, when h-BN nanolubricants were utilized, a noticeable reduction in abrasion wear was observed, resulting in a smoother surface. This reduction in wear and friction could be attributed to the formation of a protective film at the contact interface, which was facilitated by the h-BN NPs.

For future research, we intend to explore the impact of h-BN NPs in various vegetable oils at different weight fractions. Additionally, the dispersion stability of these nanolubricants will be examined, ensuring a thorough understanding of their performances and potential applications.

**Author Contributions:** Conceptualization, U.Z., S.K., R.S. and M.F.W.; methodology, U.Z., N.T., R.S. and M.F.W.; software U.Z., N.T.; validation, C.S.S., H.S.G. and S.K.; formal analysis; investigation,; resources, S.K., R.S. and M.F.W.; data curation, S.W.; writing—original draft preparation, S.K., S.W. and U.Z.; writing—review and editing, U.Z., S.K., R.S. and M.F.W.; visualization, S.K.; supervision, R.S. and M.F.W.; project administration, M.F.W.; funding acquisition, M.F.W. All authors have read and agreed to the published version of the manuscript.

**Funding:** This research received no external funding.

**Data Availability Statement:** Data Available on request.

**Conflicts of Interest:** The authors declare no conflict of interest.

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
