# Peer review of "Investigating Friction and Antiwear Characteristics of Organic and Synthetic Oils Using h-BN Nanoparticle Additives: A Tribological Study"

_lubricants, doi:10.3390/lubricants12010027_

Round 1

Reviewer 1 Report

In order to increase the service life of instruments and equipment, it is necessary to improve the performance of friction and wear. This paper aims to reduce the frictional loss of four different oils by adding hexagonal boron nitride (h-BN) nanomaterials as lubricants. Finally, the addition amount was changed to optimize the experimental results.

Here are some comments that the authors should address:

There are a number of literatures on the investigation of the addition of h-BN as a lubricant in synthetic oils. What makes this manuscript different from the others? The innovations need to be listed in the paper.

2 Maintain the same format of the pictures as much as possible. (e.g., Figure 9.)

3 The format of the references needs to be further revised and improved.

4 Additional statistical datas (e.g., standard deviations) are needed for the average wear scar diameter.

5 Standard test methods such as ASTM-D4172 need to be briefly described in the text and listed in the references.  

Some grammatical errors need to be corrected throughout the manuscript, and the language needs cleaning up.

Author Response

Response to Editor and Reviewer’s Comments

The authors would like to thank the reviewers, and editor-in-chief for the time they spent reviewing our manuscript entitled “ Investigating Friction and Antiwear Characteristics of Organic and Synthetic Oils using h-BN Nanoparticle Additives: A Tribological Study”. The suggestions provided were extremely helpful to enhance the quality of work. We have tried to answer all the questions/remarks to the best of our knowledge. The manuscript is now enriched with the incorporation of several additions and modifications based on the reviewer's comments in the revised form. You are requested to kindly peruse the answers to the reviewer’s comments.

In order to help the editor and reviewers to find the modifications introduced in the revised manuscript, the following set of color codes is used.

  • Red text: Comments of the editor and reviewers.
  • Black text: Author’s reply to the comments of the editor and reviewers.

 Yellow-shaded text: All changes and corrections were carried out in the revised manuscript.

Reviewer 1:

Comment 1:  There are a number of literatures on the investigation of the addition of h-BN as a lubricant in synthetic oils. What makes this manuscript different from the others? The innovations need to be listed in the paper.

Response: The authors agree with the concern of the reviewer regarding the novelty/ different work from others. The authors have tried to make a comparative study from the existing research findings and the same has been incorporated in the revised manuscript, section 1, Introduction.

Comment 2: Maintain the same format of the pictures as much as possible. (e.g., Figure 9.)

Response: The authors have maintained the format for all pictures as per the journal guidelines and the same has been incorporated in the revised manuscript.

Comment 3: The format of the references needs to be further revised and improved.

Response: The authors have revised and maintained all the references as per the journal guidelines and the same has been incorporated in the revised manuscript.

Comment 4: Additional statistical data (e.g., standard deviations) are needed for the average wear scar diameter.

Response: We agree with the valuable suggestion of the editor related to standard deviations for the wear scar diameter. The standard deviations have been calculated and incorporated in the revised manuscript, in section no. 4.2.1, Anti-wear properties, Fig. (10).

Comment 5: Standard test methods such as ASTM-D4172 need to be briefly described in the text and listed in the references.

Response: The test methods such as ADTM D4172 have been described and incorporated in the revised manuscript, in section no. 3, Experimental procedure.

Comment: Some grammatical errors need to be corrected throughout the manuscript, and the language needs cleaning up.

Response: The grammatical errors have been checked throughout the manuscript and have been rectified in the revised manuscript.

Reviewer 2 Report

The paper is interesting due to the topic addressed and aims to analyze the impact of hexagonal boron nitride (h-BN) nanoparticles as additives on the rheological, friction, and anti-wear characteristics of four distinct oil varieties.

However, there are still some issues to be clarified, namely:

1. Define more clearly what the purpose of the work is and how the results are validated.

2. You talk about "Compared to the base oil, ..." (line 20), which is the base oil? As I understand, after reading the paper, the four tested oils are considered basic, or not! Then, call yourself 'base oils' (see the whole work).

3. It must be clearly defined with whom the 'eco-friendly bio-lubricant' is compared (consisting of who/what?); together with the base oil (what is it?), together with 'pure oil and commercial engine oil' (what are these oils?) (see line 93-94)!

4. In line 48, isn't the boron nitride formula MoS2?

5. In line 76, "was observed for 50 N normal load at 30 mm/s and 3 wt.% of h-BN", correct '... at the speed of 30 mm/s ...'.

6. In line 78, "By comparing the friction coefficient and surface roughness of the worn surface ...", with whom did they compare?

7. You added 0.2 wt. % in the 4 types of oils, considered in the experimental tests, I ask if there would be another value of wt. % of h-BN, would similar values result, and in the same order? The value 0.2 wt. %, on what basis did you establish it, or did you consider the one from the literature to be good?

8. How was the force of 392 N reached?

9. In "Conclusions" you present some results with some comparisons and what you will do in the future, but the usefulness of this study does not emerge!

In order to improve the work (even in the conclusions), I recommend you to consult the following works with: https://doi.org/10.1177/1350650118807571 and DOI: 10.3390/lubricants4020012.

Regarding the English language, it requires a minor review of the required.

Author Response

Response to Editor and Reviewer’s Comments

The authors would like to thank the reviewers, and editor-in-chief for the time they spent reviewing our manuscript entitled “ Investigating Friction and Antiwear Characteristics of Organic and Synthetic Oils using h-BN Nanoparticle Additives: A Tribological Study”. The suggestions provided were extremely helpful to enhance the quality of work. We have tried to answer all the questions/remarks to the best of our knowledge. The manuscript is now enriched with the incorporation of several additions and modifications based on the reviewer's comments in the revised form. You are requested to kindly peruse the answers to the reviewer’s comments.

In order to help the editor and reviewers to find the modifications introduced in the revised manuscript, the following set of color codes is used.

  • Red text: Comments of the editor and reviewers.
  • Black text: Author’s reply to the comments of the editor and reviewers.

 Yellow-shaded text: All changes and corrections were carried out in the revised manuscript.

Reviewer 2:

Comment 1: Define more clearly what the purpose of the work is and how the results are validated.

Response: We agree with the concerns of the reviewer regarding the purpose of the work and the validation of results. The objective of this study is to identify the most appropriate base stock oil that exhibits low COF and antiwear properties. Currently, there are insufficient articles on the incorporation of h-BN NPs into both organic and synthetic oils. Therefore, it is crucial to investigate these novel aspects to determine the feasibility of developing nanolubricants for various engineering applications.

Comment 2: You talk about "Compared to the base oil, ..." (line 20), which is the base oil? As I understand, after reading the paper, the four tested oils are considered basic, or not! Then, call yourself 'base oils' (see the whole work).

Response: Agreed and incorporated throughout the revised manuscript.

Comment 3: It must be clearly defined with whom the 'eco-friendly bio-lubricant' is compared (consisting of who/what?); together with the base oil (what is it?), together with 'pure oil and commercial engine oil' (what are these oils?) (see line 93-94)!

Response: Agreed and corrected in the revised manuscript.

Comment 4: In line 48, isn't the boron nitride formula MoS2?

Response: We are sorry for the typos error. It has been corrected in the revised manuscript.

Comment 5: In line 76, "was observed for 50 N normal load at 30 mm/s and 3 wt.% of h-BN", correct '... at the speed of 30 mm/s ...'.

Response:  Agreed and corrected in the revised manuscript.

Comment 6: In line 78, "By comparing the friction coefficient and surface roughness of the worn surface ...", with whom did they compare?

Response: Agreed and corrected in the revised manuscript.

Comment 7: You added 0.2 wt. % in the 4 types of oils, considered in the experimental tests, I ask if there would be another value of wt. % of h-BN, would similar values result, and in the same order? The value 0.2 wt. %, on what basis did you establish it, or did you consider the one from the literature to be good?

Response: We agree with the concerns of the reviewer, at different wt. % concentration of h-BN NPs the results will be different. But in this study, the authors investigate the effect of h-BN NPs (at 0.2 wt. %) on different base oils and the concentration of 0.2 wt. % was taken from the literature. Literature studies reported that at low concentrations of h-BN NPs, the friction coefficient and antiwear properties were significantly improved.

(“DOI: 10.1080/15567036.2020.1864516”)

Comment 8: How was the force of 392 N reached?

Response: The force has been applied pneumatically which was detected by the load sensor.

Comment 9: In "Conclusions" you present some results with some comparisons and what you will do in the future, but the usefulness of this study does not emerge. In order to improve the work (even in the conclusions), I recommend you to consult the following works with: https://doi.org/10.1177/1350650118807571 and DOI: 10.3390/lubricants4020012.

Response: The conclusion section has been improved and it has been incorporated in the revised manuscript.

Reviewer 3 Report

The paper deals with a study of the friction and antiwear characteristics of organic and synthetic oils using h-BN nanoparticle additives. The impact of hexagonal boron nitride (h-BN) nanoparticles on the rheological and anti-wear characteristics of four distinct oil varieties, namely SAE20W50, Soybean, PAO-4, and Olive oils was studied. Some useful research results have been obtained. In general, the research content of this paper is rich. However, some areas need to be modified before final acceptance.

1. The language expression needs to be carefully polished with the help of English professionals or professional organizations.

2. The tests parameter of “Load-392” needs to be carefully checked and confirmed.

3. The friction torque curves in Figures 6a and b are illegible, please redraw them.

4. How the values for the bar chart in Figure 6 are derived. How were the COF values in Figure 7 obtained, and how were they calculated?

5. Optical images of wear scars of balls are poor, don't you even have a ruler?

The language expression needs to be carefully polished with the help of English professionals or professional organizations.

Author Response

Response to Editor and Reviewer’s Comments

The authors would like to thank the reviewers, and editor-in-chief for the time they spent reviewing our manuscript entitled “ Investigating Friction and Antiwear Characteristics of Organic and Synthetic Oils using h-BN Nanoparticle Additives: A Tribological Study”. The suggestions provided were extremely helpful to enhance the quality of work. We have tried to answer all the questions/remarks to the best of our knowledge. The manuscript is now enriched with the incorporation of several additions and modifications based on the reviewer's comments in the revised form. You are requested to kindly peruse the answers to the reviewer’s comments.

In order to help the editor and reviewers to find the modifications introduced in the revised manuscript, the following set of color codes is used.

  • Red text: Comments of the editor and reviewers.
  • Black text: Author’s reply to the comments of the editor and reviewers.

 Yellow-shaded text: All changes and corrections were carried out in the revised manuscript.

Reviewer 3:

Comment 1: The language expression needs to be carefully polished with the help of English professionals or professional organizations.

Response: The grammatical errors have been checked throughout the manuscript and the English language has been polished through in the revised manuscript.

Comment 2: The tests parameter of “Load-392” needs to be carefully checked and confirmed.

Response: The test parameter such as load = 392N, time = 3600 seconds, and speed of spindle = 1200 RPM were selected as per the ASTM D4172.

Comment 3: The friction torque curves in Figures 6a and b are illegible, please redraw them.

Response: Figure 6a and 6b has been redrawn and the same has been incorporated in the revised manuscript.

Comment 4: How the values for the bar chart in Figure 6 are derived. How were the COF values in Figure 7 obtained, and how were they calculated?

Response: The value of COF has been calculated by recording the frictional torque. The following equation has been used for obtaining the COF and the same has been incorporated in the revised manuscript.

Where µ is COF, T is the torque of the upper shaft (kg-mm), r = 3.6 mm (distance from the axis of rotation to the center of the contact of the fixed ball) and W is load (kg).

Comment 5: Optical images of wear scars of balls are poor, don't you even have a ruler?

Response: The optical image of the wear scar has been changed and incorporated in the revised manuscript.

Round 2

Reviewer 1 Report

The revision was enough for publication.

Reviewer 2 Report

The authors took into account the reviewer's suggestions and observations. As a result, the work was greatly improved by additions and corrections.

Reviewer 3 Report

The authors have improved some problems and the paper is acceptable.